# Antifouling Lipids from Marine Fungi of the Beibu Gulf

**DOI:** 10.3390/metabo15110721

**Published:** 2025-11-05

**Authors:** Mengfan Qi, Wang Jiang, Huaqing Huang, Lu Lu, Zhiwei Su, Xiaowei Luo, Chenghai Gao, Yonghong Liu, Xinya Xu

**Affiliations:** 1Institute of Marine Drugs, Guangxi Key Laboratory of Marine Drugs, Guangxi University of Chinese Medicine, Nanning 530200, China; 2University Engineering Research Center of High-Efficient Utilization of Marine Traditional Chinese Medicine Resources, Guangxi, Nanning 530200, China; 3College of Agriculture, Guangxi University, Nanning 530004, China

**Keywords:** marine fungi, lipid, biofouling, antifouling, barnacles, natural products

## Abstract

**Background**: The search for environmentally friendly antifouling agents has led to an increased focus on marine natural products. **Methods**: This study investigated the antifouling potential of lipid fractions extracted from ten marine fungal strains isolated from the Beibu Gulf, China. The lipids were evaluated through a multi-level bioassay approach, including the inhibition of microfouling (against four fouling bacteria: *Marinobacterium jannaschii*, *Vibrio pelagius*, *Vibrio rotiferianus*, and *Alteromonas macleodii*), the prevention of macrofouling (inhibition of barnacle *Amphibalanus reticulatus* cyprid settlement), and long-term (90-day) marine field trials. **Results**: Eight lipid fractions demonstrated inhibitory effects against at least one bacterial strain. Five lipids significantly inhibited barnacle cyprid settlement, with half-maximal effective concentration (EC_50_) values ranging from 0.21 to 1.81 µg/mL and exhibited low toxicity (half-maximal lethal concentration (LC_50_) > 50 µg/mL). Notably, four lipid fractions maintained potent antifouling efficacy (>70% inhibition) throughout the 90-day field exposure. Chemical characterization via gas chromatography–mass spectrometry (GC–MS) revealed that the bioactive fractions were predominantly composed of fatty acids and their derivatives. Major identified compounds included palmitic acid, methyl palmitate, linoleic acid, dodecyl-9-ynyl chloroacetate, *cis*-13-octadecenoic acid, oleic acid, methyl 11,14-octadecadienoate, and (*E*)-9-octadecenoic acid methyl ester. **Conclusions**: This work represents the first comprehensive investigation of marine fungal lipids from the Beibu Gulf with multi-target antifouling properties, providing a theoretical foundation and practical candidate compounds for developing eco-friendly antifouling coatings.

## 1. Introduction

Marine biofouling, refers to the settlement of fouling organisms on submerged surfaces, poses significant economic and ecological threats to maritime industries and marine ecosystems [1,2]. The fouling process initiates with the adsorption of microorganisms and the subsequent formation of biofilm, a complex matrix of proteins and polysaccharides, on submerged surfaces. This biofilm facilitates the attachment of microalgae and invertebrate larvae within hours to days, culminating in a mature and stable fouling community within 2–3 weeks [3]. The resulting increase in hull roughness and drag leads to an estimated 8–15% rise in fuel consumption for commercial vessels, contributing to annual global economic losses surpassing $150 billion and significantly increasing greenhouse gas emissions [4,5]. Historically, heavy metal-based coatings, particularly those containing tributyltin (TBT), were widely used for their cost-effectiveness and efficacy. However, the persistence and toxicity of TBT in the marine environment led to severe ecological consequences, including bioaccumulation in seafood and impacts on non-target organisms, prompting a global ban by the International Maritime Organization (IMO) in 2008 [6,7]. This regulatory shift has driven urgent demand for non-toxic, biodegradable, and eco-friendly antifouling alternatives [8,9].

Marine organisms represent a prolific source of structurally diverse bioactive secondary metabolites with potential applications in antifouling technology [10]. As key participants in marine biogeochemical cycles, fungi are widely distributed throughout the ocean, occurring from the surface waters to sediments thousands of meters deep, and exert influence on other marine organisms [11]. Marine fungi have also gained increasing attention for their metabolic versatility and the ecological promise of their bioactive compounds [12]. For example, the marine fungus *Aspergillus westerdijkiae* DFFSCS013, isolated from sediments in the South China Sea at a depth exceeding 2000 m, with alkaloids as its major metabolites, has been shown to inhibit fouling bacteria as well as the fouling organism *Bugula neritina*. Furthermore, field experiments have demonstrated its long-lasting anti-fouling efficacy [13].

Natural products derived from marine sources offer distinct advantages, such as environmental compatibility, low ecotoxicity, and biodegradability [14]. Various compound classes, including fatty acids, phenolic derivatives, diphenyl ethers, and alkaloids, have demonstrated promising antifouling activity [15]. For instance, bacterially derived fatty acids like 2-hydroxytetradecanoic acid and *cis*-9-oleic acid from *Shewanella oneidensis* showed activity against fouling bacteria [16]. Similarly, synthetic *n*-alkyl 2-furoates incorporated into coatings exhibited strong antifouling performance [17]. Despite these advances, comprehensive evaluations of lipid components, particularly from marine fungi, against multi-scale fouling (encompassing microfouling, macrofouling, and long-term field performance) remain scarce. As our ongoing search for bioactive natural metabolites from marine fungi, ten fungal lipid fractions were obtained and determined for their antifouling effects. This study presents the first comprehensive screening of marine fungal lipids for their multifaceted antifouling performance, including antibacterial assays, barnacle cyprid settlement inhibition tests, and extended marine field trials. Furthermore, the principal chemical constituents of the most active fractions were analyzed using GC–MS. Our findings highlight the potential of marine fungal lipids as a sustainable source for developing natural antifouling agents.

## 2. Materials and Methods

### 2.1. Fungal Material and Lipid Extraction

Ten marine fungal strains were isolated from various substrates, including seawater, salt field sediments, sponges, and corals, collected from the Beibu Gulf, China (Table 1, Figure 1). The strains were identified by its morphological features (Figure 2) and the sequence analysis of the internally transcribed spacer (ITS) region of the rRNA gene (Appendix A). All the strains were preserved at the Institute of Marine Medicine, Guangxi University of Chinese Medicine. After activation on potato dextrose agar (PDA) plates, each strain was inoculated into 1 L Erlenmeyer flasks containing 400 mL of fermentation medium (mannitol 2%, yeast extract 0.3%, MgSO_4_ 0.03%, sodium glutamate 1%, glucose 1%, KH_2_PO_4_ 0.05%, maltose 2%, sea salt 3%) and incubated at 28 °C for 30 days under static conditions. The entire fermentation broth of each strain was extracted three times with an equal volume of petroleum ether (60–90 °C). The combined organic extracts were concentrated in vacuo at 40 °C to obtain the lipid fractions, which were stored at 4 °C until further use.

### 2.2. Anti-Fouling Bacterial Activity Assay

The inhibitory activities of lipid components against fouling bacteria were determined by the filter paper method [18]. The fouling bacteria *Marinobacterium jannaschii* (MJ), *Vibrio Pelagius* (VP), *V. rotiferianus* (VR) and *Alteromonas macleodii* (AM) were isolated from polyvinyl chloride (PVC) panels settled by marine fouling organisms, kindly donated by Dr. Zhang Xiaoyong, South China Agricultural University. Each bacterial strain was cultured in LB medium at 30 °C with shaking at 180 rpm until the optical density at 600 nm (OD_600_) reached 0.6–0.8 (logarithmic phase). A 100 μL aliquot of the bacterial suspension was spread evenly onto LB agar plates. Sterile 6 mm diameter filter paper discs were placed on the inoculated agar surfaces. A 5 μL volume of each lipid fraction, dissolved in dimethyl sulfoxide (DMSO) at a concentration of 10 mg/mL, was applied to the discs. Discs containing only DMSO served as the negative control, penicillin and chloramphenicol (10 μg/disc) were used as the positive control. All assays were performed in triplicate. The plates were incubated at 37 °C for 24 h, after which the diameters of the inhibition zones (including the disc) were measured in millimeters.

### 2.3. Anti-Barnacle Cyprid Larval Settlement Assay

The anti-larval settlement activity was assessed against cyprid larvae of the barnacle *Amphibalanus reticulatus* according to a previously described method [19] with minor modifications. Adult barnacles were collected from the coastal waters of Fangchenggang, Guangxi, and identified by Dr. Liu Xinming (Institute of Marine Medicine, Guangxi University of Chinese Medicine). Adults were maintained in filtered, aged seawater and fed the diatom *Phaeodactylum tricornutum*. Nauplii larvae were collected using light attraction and reared to the cyprid stage on a diet of *P. tricornutum* over 5–7 days. For the bioassay, 995 μL of filtered seawater, containing 10–15 actively swimming cyprids, was added to each well of a 24-well plate. Then, 5 μL of the test sample (dissolved in DMSO) was added to achieve the desired final concentration. The solution was gently mixed, and the plates were incubated at room temperature (25 °C) under a natural light cycle for 24 h. Each concentration was tested in triplicate. DMSO (0.5% *v*/*v*) served as the negative control, and the commercial antifoulant SeaNine-211 was used as the positive control. After incubation, the number of settled (attached) and dead (non-motile) cyprids in each well was counted under a stereomicroscope. The EC_50_ (concentration for 50% settlement inhibition) and LC_50_ (concentration causing 50% mortality) values were calculated using probit analysis.

### 2.4. Antifouling Marine Field Trial

Marine field trials were conducted from March to June 2024 in a natural harbor environment in the Beibu Gulf. PVC panels (10 cm × 10 cm) were used as substrates. Approximately 5 mL of a commercial priming paint was applied to each panel and allowed to dry. For the test groups, 400 mg of each liquid lipid fraction was thoroughly mixed with the priming paint before application to the panels. Panels coated with priming paint containing cuprous oxide served as the positive control, while panels with priming paint alone served as the blank control. All panels were submerged in seawater at a depth of approximately 1 m. Fouling progression was monitored and recorded at 15, 30, 45, 60, and 90 days. The percentage of surface area covered by fouling organisms (e.g., bryozoans, tube worms, algae) was visually estimated for each panel [20]. The antifouling efficacy was calculated based on the reduction in fouling coverage compared to the blank control.

### 2.5. GC-MS Analysis

The chemical composition of the selected lipid fractions was analyzed by GC–MS. A 20 μL aliquot of each lipid sample was dissolved in 980 μL of hexane. Subsequently, 50 μL of 2 N potassium hydroxide in methanol (prepared by dissolving 11.2 g of KOH in 100 mL of methanol) was added for derivatization. The mixture was vortexed, sonicated for 10 min, and then centrifuged at 12,000 rpm for 10 min. The supernatant was carefully transferred to a GC vial for analysis. GC–MS analysis was performed using an Agilent 7890B GC system coupled with an Agilent 5977A MSD (Agilent Technologies, Santa Clara, CA, USA). Separation was achieved on an HP-5MS capillary column (30 m × 250 μm × 0.25 μm). High-purity helium was used as the carrier gas at a constant flow rate of 1.0 mL/min. The injection volume was 1 μL in splitless mode, with the injector temperature set at 250 °C. The oven temperature program was as follows: initial temperature 100 °C (held for 2 min), ramped to 180 °C at 3 °C/min (held for 1 min), then ramped to 250 °C at 10 °C/min (held for 5 min). The total run time was 35.33 min. The mass spectrometer was operated in electron ionization (EI) mode at 70 eV. The ion source temperature was 230 °C, and the quadrupole temperature was 150 °C. The mass scan range was *m*/*z* 50–550. Solvent delay was set at 3 min. Compound identification was tentatively assigned by comparing the mass spectra with the NIST 14 and Wiley database.

### 2.6. Statistical Analysis

All laboratory bioassays were conducted with at least three independent replicates. Data are presented as mean ± standard deviation (SD). The EC_50_ and LC_50_ values for the barnacle assay were calculated using probit regression analysis in SPSS software (version 26.0, IBM Corp., Armonk, NY, USA).

### 2.7. Ethics Statement

The barnacles (*Amphibalanus reticulatus*) used in this study are invertebrates and are not subject to ethical approval requirements according to national and institutional guidelines. All efforts were made to minimize suffering during collection and maintenance.

## 3. Results

### 3.1. Anti-Fouling Bacterial Activity

The antibacterial activities of the ten fungal lipid fractions against four fouling bacterial strains are summarized in Table 2. Eight out of the ten lipid fractions (80%) exhibited inhibitory activity against at least one bacterial strain. The lipid fraction from strain GXIMD00545 (*Curvularia lunata*) showed the broadest spectrum of activity, inhibiting all four indicator bacteria, with inhibition zones ranging from 7.15 ± 0.33 mm (against MJ) to 8.08 ± 0.37 mm (against VR). The lipid from GXIMD00543 (*Aspergillus carneus*) was active against three strains (VR, MJ, AM). Strains GXIMD00519 and GXIMD00548 were active against two bacterial strains each.

### 3.2. Anti-Barnacle Cyprid Larval Settlement Activity

The results of the barnacle cyprid settlement inhibition assay are presented in Table 3. Seven lipid fractions exhibited anti-settlement activity. Four lipid fractions, GXIMD00527, GXIMD00548, GXIMD00519, and GXIMD00541, showed particularly high potency, with EC_50_ values of 0.23, 0.21, 0.59, and 1.81 μg/mL, respectively. Importantly, these fractions demonstrated low toxicity, with LC_50_ values exceeding 50 μg/mL, resulting in high therapeutic ratios (LC_50_/EC_50_). The activity of these three fractions was superior to the positive control, SeaNine-211 (EC_50_ = 1.95 μg/mL). The lipid fractions from GXIMD00547 and GXIMD00543 also showed strong activity, with EC_50_ values of 3.89 and 5.50 μg/mL, respectively, which were comparable to or better than the positive control.

### 3.3. Antifouling Marine Field Trial

The results of the 90-day marine field trial are summarized in Table 4 and illustrated in Figure 3 and Appendix A. The blank control panels (priming paint only) were rapidly colonized, showing the first signs of bryozoans and tube worms within 15 days and reaching 100% fouling coverage by day 90. In contrast, the positive control panels (cuprous oxide coating) showed significantly reduced fouling, with a final coverage of 14% at 90 days. The lipid fractions demonstrated varying degrees of long-term efficacy. The coatings containing GXIMD00527 and GXIMD00547 lipids exhibited excellent and sustained antifouling performance, with fouling coverage remaining below 25% throughout the entire trial period, ultimately reaching 14% and 18% at day 90, respectively. This performance was comparable to the cuprous oxide positive control. The lipid fractions GXIMD00548 and GXIMD00541 also showed significant activity, with final fouling coverages of 28% and 47%, respectively, which were substantially lower than the blank control.

### 3.4. Chemical Composition of Selected Lipids Fractions

To identify the compounds responsible for the observed antifouling activity, the three most promising lipid fractions (GXIMD00527, GXIMD00548, and GXIMD00543) were subjected to GC–MS analysis. The relative contents of the identified lipids are listed in Table 5. The lipid fraction from GXIMD00527 was complex, containing 14 identified compounds accounting for 99.46% of the total volatile content. It was dominated by dodecyl-9-alkynyl chloroacetate (34.80%), palmitic acid (28.35%), and *cis*-13-octadecenoic acid (15.11%) (Figure 4). In contrast, GXIMD00548 contained seven identified compounds (95.56% of total volatiles), with linoleic acid being overwhelmingly predominant (67.48%), followed by palmitic acid (19.59%). The GXIMD00543 fraction contained 15 identified compounds (99.60% of total volatiles), with methyl 11,14-octadecadienoate (28.17%), methyl palmitate (19.35%), and dodecyl-9-alkynyl chloroacetate (17.19%) as the major constituents. Notably, palmitic acid and methyl palmitate were present in all three bioactive fractions. Dodecyl-9-alkynyl chloroacetate was a major component in both GXIMD00527 and GXIMD00543.

## 4. Discussion

The search for sustainable and eco-friendly antifouling solutions has intensified following the global ban on TBT-based coatings. Marine fungi, with their vast and largely untapped metabolic diversity, represent a promising source of novel bioactive compounds [12,21]. In this study, we demonstrated that lipid fractions from marine fungi isolated from the Beibu Gulf possess significant and multi-faceted antifouling activity.

The antibacterial screening revealed that 80% of the tested lipid fractions were active against at least one of the four common marine fouling bacteria, indicating a broad-spectrum potential for preventing the initial stages of biofilm formation, which is critical for the subsequent recruitment of macrofoulers [3]. The lipid from *Curvularia lunata* (GXIMD00545) was particularly notable for its activity against all four test strains.

More impressively, in the anti-settlement assay against barnacle cyprids, several lipid fractions exhibited remarkable potency and selectivity. The EC_50_ values for GXIMD00527 (0.23 μg/mL), GXIMD00548 (0.21 μg/mL), and GXIMD00519 (0.59 μg/mL) were lower than that of the commercial standard SeaNine-211 (1.95 μg/mL), highlighting their exceptional efficacy. Furthermore, the lack of observed toxicity (LC_50_ > 50 μg/mL) at the effective concentrations is a crucial advantage for environmentally benign antifouling agents, as it suggests a non-biocidal mode of action, possibly through the inhibition of larval settlement signaling pathways or temporary sensory manipulation [22]. SeaNine 211, as one of the most widely used small-molecule antifoulants, is generally considered to have a favorable safety profile. However, compared to the lipid fractions mentioned above, it has a relatively narrow LC_50_ range (33.93 μg/mL). Literature reports indicate that SeaNine 211 can exert long-term effects on marine phytoplankton communities, exhibit immunotoxicity to the hemocytes of edible bivalve mollusks, and potentially cause adverse effects on sea urchins, suggesting a potential risk of global environmental pollution [23,24].

The long-term marine field trial provided the most compelling evidence for the practical application of these lipid fractions. The sustained efficacy of GXIMD00527 and GXIMD00547 over 90 days, with fouling coverage remaining comparable to the cuprous oxide control, underscores their potential as active ingredients in coatings. The fluctuating settlement rate observed for GXIMD00527 (e.g., 34% at 30 days dropping to 15% at 45 days) could be related to dynamic environmental factors, leaching rates, or the succession of fouling communities, which is a common phenomenon in field tests [20].

The GC–MS analysis of the most active fractions (GXIMD00527, GXIMD00548, GXIMD00543) aimed to identify the key metabolites responsible for the observed activities. The consistent presence of palmitic acid and methyl palmitate in all three fractions is significant. Previous studies have reported the antifouling properties of fatty acids. For instance, Gao et al. [25] identified palmitic acid, myristic acid, and octadecanoic acid as the active principles in antifouling bacteria. Fatty acids can disrupt fouling bacterial quorum sensing and interfere with the settlement and metamorphosis of invertebrate larvae [16,26]. The high concentration of linoleic acid in GXIMD00548 (67.48%) suggests it may be a key active component in this specific fraction, contributing to its strong anti-settlement activity. Compound dodecyl-9-alkynyl chloroacetate was a major component in both GXIMD00527 (34.80%) and GXIMD00543 (17.19%). This compound, containing both an alkyne and a chloroacetate functional group, is a relatively unusual natural product and may contribute significantly to the potent and broad-spectrum activity, especially of GXIMD00527. Its mode of action warrants further investigation. The variation in composition among the active fractions suggests that antifouling efficacy may not be attributable to a single compound but could result from synergistic interactions within the lipid mixture [27]. This synergy could potentially broaden the spectrum of activity and delay the development of resistance in fouling organisms.

While natural products offer great promise, challenges such as limited natural abundance, compound stability, and cost-effective production remain [28]. The high relative content of the identified active compounds (e.g., palmitic acid, dodecyl-9-alkynyl chloroacetate) in these fungal lipids is advantageous. Future work should focus on the sustainable production of these compounds, possibly through optimized fermentation of the fungal strains or synthesis of the most active analogues, to ensure a viable supply for coating formulations.

## 5. Conclusions

This study provides comprehensive evidence for the antifouling potential of lipid fractions derived from marine fungi of the Beibu Gulf, China. Through a multi-tiered evaluation strategy, we identified several lipid fractions, particularly from strains GXIMD00527, GXIMD00547, and GXIMD00548, that exhibit strong activity against both microfouling (bacteria) and macrofouling (barnacle settlement), while maintaining efficacy in long-term field trials. GC–MS analysis indicated that fatty acids such as palmitic acid and their esters, along with unusual compounds like dodecyl-9-alkynyl chloroacetate, are likely key contributors to this activity.

These findings not only validate the role of marine fungi as a valuable resource for green antifouling technologies but also identify specific candidate compounds and producing strains for further development. Future research will focus on elucidating the precise molecular mechanisms of action of these lipid components, optimizing their production, and incorporating them into practical, durable, and environmentally friendly coating systems.

## Figures and Tables

**Figure 1 metabolites-15-00721-f001:**
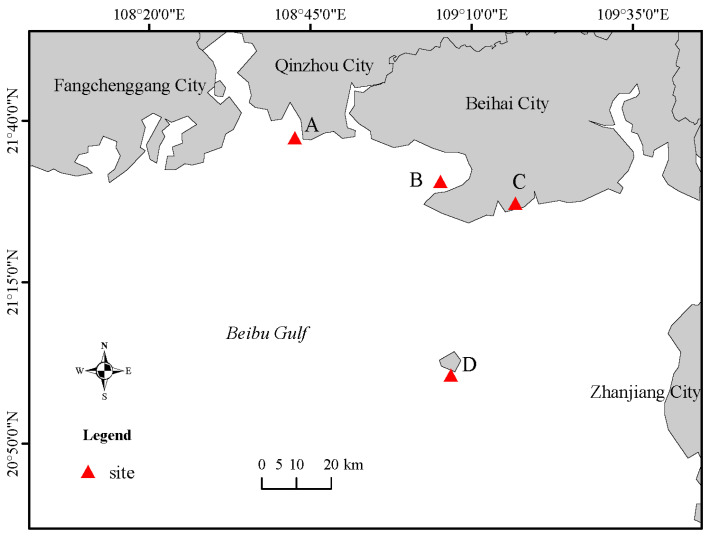
Map of Beibu gulf and the location of the sampling sites (A: Qinzhou seawater; B: Beihai seawater; C: Beihai Zhulin salt field; D: Weizhou Island).

**Figure 2 metabolites-15-00721-f002:**
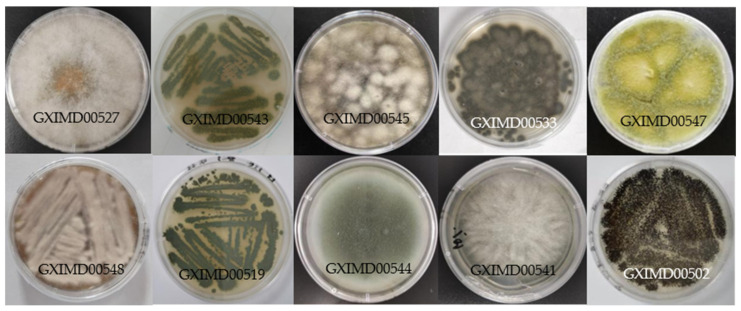
The colony morphology images of 10 strains marine fungi.

**Figure 3 metabolites-15-00721-f003:**
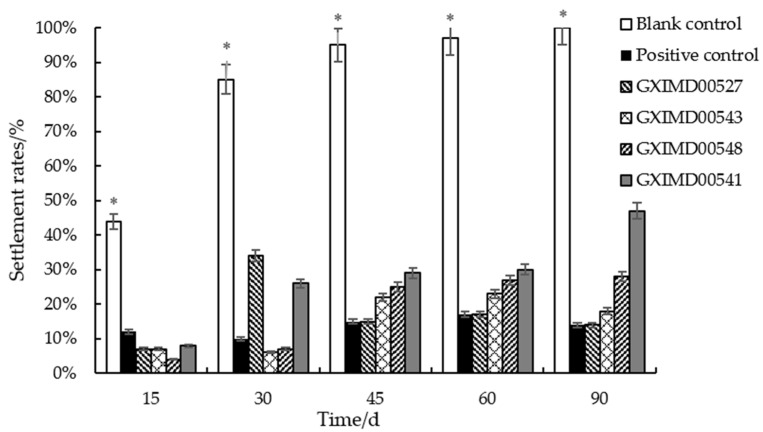
Biofouling settlement rates on PVC panels coated with blank control, positive control, and selected lipid fractions (GXIMD00527, GXIMD00543, GXIMD00548, GXIMD00541) over a 90-day marine field trial. * showed that there was a significant difference between the experimental group and the other groups (*p* < 0.05).

**Figure 4 metabolites-15-00721-f004:**
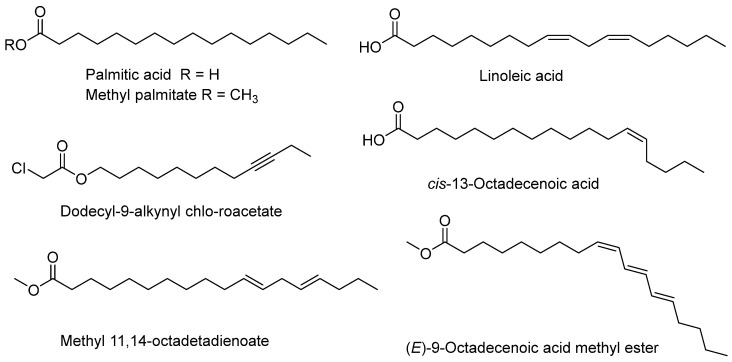
The main chemical structures of lipids (relative contents more than 10%).

**Table 1 metabolites-15-00721-t001:** The origin and species of the ten marine fungal strains.

No.	Strain Number	Species	Source
1	GXIMD00527	*Fusarium incarnatum*	Beihai Zhulin salt field
2	GXIMD00543	*Aspergillus carneus*	Sponge, Weizhou Island, Beihai
3	GXIMD00545	*Curvularia lunata*	Beihai seawater
4	GXIMD00533	*Cladosporium cladosporioides*	Beihai Zhulin salt field
5	GXIMD00547	*Trichoderma brevicompactum*	Qinzhou seawater
6	GXIMD00548	*Aspergillus iizukae*	Beihai Zhulin salt field
7	GXIMD00519	*Aspergillus carneus*	Coral, Weizhou Island, Beihai
8	GXIMD00544	*Aspergillus fumigatus*	Beihai Zhulin salt field
9	GXIMD00541	*Fusarium chlamydosporum*	Beihai Zhulin salt field
10	GXIMD00502	*Aspergillus niger*	Coral, Weizhou Island, Beihai

**Table 2 metabolites-15-00721-t002:** Antibacterial activity of ten marine fungal lipid fractions (inhibition zone diameter in mm) ^a^.

Fungal Lipid Fraction	Strain Number	VR	VP	MJ	AM
1	GXIMD00519	8.19 ± 0.75	— ^b^	—	6.90 ± 0.54
2	GXIMD00541	7.00 ± 0.22	—	—	—
3	GXIMD00545	8.08 ± 0.37	7.21 ± 0.11	7.15 ± 0.33	7.99 ± 0.30
4	GXIMD00502	—	6.47 ± 0.23	—	—
5	GXIMD00547	—	—	—	—
6	GXIMD00543	7.58 ± 0.71	—	7.73 ± 0.21	6.66 ± 0.38
7	GXIMD00544	—	7.62 ± 0.22	—	—
8	GXIMD00527	—	6.84 ± 0.02	—	—
9	GXIMD00548	—	—	7.24 ± 0.12	6.34 ± 0.06
10	GXIMD00533	—	—	—	—
Positive control	Penicillin	18.16 ± 0.11	12.45 ± 0.22	17.18 ± 0.04	9.47 ± 0.07
	Chloramphenicol	14.88 ± 0.05	—	16.23 ± 0.14	—

^a^ Values represent mean ± standard deviation (*n* = 3). ^b^ — indicates no observable inhibition zone.

**Table 3 metabolites-15-00721-t003:** Anti-settlement activity (EC_50_) and toxicity (LC_50_) of marine fungal lipid fractions against barnacle cyprids.

Strain Number	EC_50_ (μg/mL) ^a^	LC_50_ (μg/mL) ^a^
GXIMD00543	5.50 ± 0.25	>50
GXIMD00541	1.81 ± 0.23	>50
GXIMD00533	13.92 ± 0.41	>50
GXIMD00527	0.23 ± 0.13	>50
GXIMD00548	0.21 ± 0.13	>50
GXIMD00519	0.59 ± 0.27	50
GXIMD00547	3.89 ± 0.32	>50
seaNine-211	1.95 ± 0.05	33.93 ± 2.06

^a^ Values represent mean ± standard deviation (*n* = 3), calculated by probit analysis.

**Table 4 metabolites-15-00721-t004:** Biofouling settlement rates (%) on PVC panels coated with selected lipid fractions during the 90-day marine field trial.

Sample Number	Settlement Rates (%)				
	15 d	30 d	45 d	60 d	90 d
GXIMD00543	7	6	22	23	18
GXIMD00541	8	26	29	30	47
GXIMD00533	27	48	67	68	49
GXIMD00527	7	43	15	17	20
GXIMD00548	4	7	25	27	28
GXIMD00519	9	15	23	19	72
GXIMD00547	10	17	55	60	57
Positive control	12	10	15	17	14
Blank control	44	85	95	97	100

**Table 5 metabolites-15-00721-t005:** Chemical composition and relative content (%) of volatile components in selected lipid from strains GXIMD00527, GXIMD00548, and GXIMD00543.

	Compound	Molecular Formula	Relative Content (%)		
			GXIMD00527	GXIMD00548	GXIMD00543
1	2,4-Di-tert-butylphenol	C_14_H_22_O	— ^a^	1.47	0.31
2	Methyl palmitate	C_17_H_34_O_2_	2.73	2.33	19.35
3	Palmitic acid	C_16_H_32_O_2_	28.35	19.59	7.9
4	Methyl 9,10-octadecadienoate	C_19_H_34_O_2_	—	2.65	—
5	Methyl 9-octadecenoate	C_19_H_36_O_2_	1.38	1.77	—
6	Methyl 8-methyl nonanoate	C_11_H_22_O_2_	—	1.27	—
7	Linoleic acid	C_18_H_32_O_2_	—	67.48	—
8	2,5-Di-tert-butylphenol	C_14_H_22_O	1.12	—	—
9	Ethyl palmitate	C_18_H_36_O_2_	0.6	—	—
10	Methyl linoleate	C_19_H_34_O_2_	1.7	—	—
11	Methyl stearate	C_19_H_38_O_2_	1.43	—	3.67
12	Dodecyl-9-alkynyl chloroacetate	C_14_H_23_ClO_2_	34.8	—	17.19
13	*cis*-13-Octadecenoic acid	C_18_H_34_O_2_	15.11	—	—
14	Stearic acid	C_18_H_36_O_2_	8.96	—	0.86
15	Butyl palmitate	C_20_H_40_O_2_	1.5	—	0.72
16	2-Chloroethyl linoleate	C_20_H_35_ClO_2_	0.72	—	—
17	2,2,2-Trifluoroethanol	C_20_H_35_F_3_O_2_	0.52	—	—
18	Butyl octadecanoate	C_22_H_44_O_2_	0.54	—	—
19	Oleic acid	C_18_H_34_O_2_	—	—	3.66
20	Methyl 12-methyltridecanoate	C_15_H_30_O_2_	—	—	0.76
21	Methyl pentadecanoate	C_16_H_32_O_2_	—	—	0.31
22	Methyl 11,14-octadetadienoate	C_19_H_34_O_2_	—	—	28.17
23	(*E*)-9-Octadecenoic acid methyl ester	C_19_H_36_O_2_	—	—	14.5
24	Phenazocine	C_22_H_27_NO	—	—	1.02
25	Ethyl linoleate	C_20_H_36_O_2_	—	—	0.77
26	2-Hydroxy-1-(hydroxymethyl)ethyl ester cetanoate	C_19_H_38_O_4_	—	—	0.41
	Total identified		99.46	95.56	99.60

^a^ —: Not detected.

## Data Availability

The original data presented in the study are included in the article/Appendix A. Further inquiries can be directed to the corresponding author.

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
