# Peer review of "Antifouling Lipids from Marine Fungi of the Beibu Gulf"

_metabolites, 2025, doi:10.3390/metabo15110721_

Round 1
Reviewer 1 Report
Comments and Suggestions for Authors
Lipids are main metabolic components of microorganisms, yet their biological functions and applications have long been overlooked. This manuscript systematically investigated the antifouling activities of lipid components derived from 10 strains of marine fungi. Some of these lipids demonstrated remarkable antibacterial efficacy against fouling bacteria and inhibited barnacle larval settlement, while also exhibiting long-term antifouling effects in field experiments. Additionally, the authors elucidated the primary chemical constituents of these lipids. These findings will contribute to the development of novel natural antifouling agents. Thus, this manuscript can be accepted after addressing the following issues.
Please provide the chemical structures of the major lipid components.
Please add the sampling map for seawater, salt field sediments, sponges, and corals from the Beibu Gulf, China.
The toxicity (LCâ‚…â‚€) of marine fungal lipid fractions against barnacle cyprids were more than 50 ug/mL in Table 3. Can you please provide the detailed values of LC50?
Author Response
Comment 1: Lipids are main metabolic components of microorganisms, yet their biological functions and applications have long been overlooked. This manuscript systematically investigated the antifouling activities of lipid components derived from 10 strains of marine fungi. Some of these lipids demonstrated remarkable antibacterial efficacy against fouling bacteria and inhibited barnacle larval settlement, while also exhibiting long-term antifouling effects in field experiments. Additionally, the authors elucidated the primary chemical constituents of these lipids. These findings will contribute to the development of novel natural antifouling agents. Thus, this manuscript can be accepted after addressing the following issues.
Response: Thank you for your warm comments.
Comment 2: Please provide the chemical structures of the major lipid components.
Response: The chemical structures of lipids with relative contents more than 10% were supplemented as Figure 4.
Comment 3: Please add the sampling map for seawater, salt field sediments, sponges, and corals from the Beibu Gulf, China.
Response: The sampling map of seawater, salt field sediments, sponges, and corals from the Beibu Gulf was supplemented as Figure 1.
Comment 4: The toxicity (LCâ‚…â‚€) of marine fungal lipid fractions against barnacle cyprids were more than 50 ug/mL in Table 3. Can you please provide the detailed values of LC50?
Response: In general, a compound is considered as non-toxic antifoulant if its LCâ‚…â‚€ > 25 µg/mL and its LCâ‚…â‚€/ECâ‚…â‚€ ratio > 15 [1]. Therefore, to ensure a clear safety margin, we set the LCâ‚…â‚€ concentration for the anti-settlement assay of fungal lipid fractions at 50 µg/mL, a value well above the required minimum. Fortunately, all determined lipids exhibited LCâ‚…â‚€ value greater than 50 µg/mL.
- Liu, L.L.; Wu, C.H.; Qian, P.Y. Marine natural products as antifouling molecules – a mini-review (2014–2020). Biofouling 2020, 36, 1210-1226.
Reviewer 2 Report
Comments and Suggestions for Authors
The manuscript entitled “Antifouling lipids from marine fungi of the Beibu Gulf” focuses on the effectiveness of ten marine fungal strains in preventing the development of micro- and macrofouling, considering a 90-day evaluation period. The overall quality of the manuscript is outstanding, it is well-structured, scientifically sound, and addresses a highly relevant and timely topic concerning the use of bioactive substances as environmentally friendly alternatives to conventional antifouling chemicals used in boats and harbors.
Although this is an excellent paper, I would like to raise a few minor comments for the authors’ consideration after carefully going through the entire manuscript.
Abstract
The abstract provides an accurate and concise summary of the study, allowing readers to grasp its main objectives and findings. I recommend, however, that the authors modify one of the keywords and include “lipid” as one of them, to better reflect the study’s focus.
Introduction
The introduction is well-written and logically organized. Nevertheless, it would benefit from a short background paragraph on marine fungi and their ecological roles in the marine environment, to better contextualize their potential as sources of antifouling compounds.
Materials and Methods
The methods are clearly described, making it easy to follow the experimental procedures. However, in the GC–MS analysis, the confirmation of some compounds using analytical standards would strengthen the validity of the chemical identifications.
Results
The results are well presented and focused on the most relevant findings.
-
In Table 2, the first column is missing a heading; I suggest labeling it as “Fungal lipid fraction” to clarify the distinction among the ten tested fractions.
-
In line 177, there seems to be a minor error — the authors refer to four lipid extractions instead of three.
Apart from these minor issues and a few typographical errors, the results section is clearly written and easy to follow.
Discussion
The discussion would benefit from a deeper analysis of some key aspects. I recommend expanding this section by incorporating more comparisons with existing literature on antifouling compounds derived from marine microorganisms. This would help strengthen the interpretation of the findings and highlight the novelty of the work.
Conclusions
The conclusions are well supported by the presented results and appropriately summarize the study’s main contributions.
Additional comments
-
Have the authors explored, or do they plan to explore, the molecular mechanisms underlying the antifouling activity of the identified lipids?
-
Given that long-term stability and resistance remain major challenges for antifouling compounds, how do the authors envision addressing the durability of these lipid-based antifouling agents over extended periods (e.g., several years)?
In view of the manuscript’s high quality, clarity, and scientific relevance, I recommend it for minor revisions before acceptance.
Author Response
Comment 1: The manuscript entitled “Antifouling lipids from marine fungi of the Beibu Gulf” focuses on the effectiveness of ten marine fungal strains in preventing the development of micro- and macrofouling, considering a 90-day evaluation period. The overall quality of the manuscript is outstanding, it is well-structured, scientifically sound, and addresses a highly relevant and timely topic concerning the use of bioactive substances as environmentally friendly alternatives to conventional antifouling chemicals used in boats and harbors.
Although this is an excellent paper, I would like to raise a few minor comments for the authors’ consideration after carefully going through the entire manuscript.
Response: Thank you for your kind words.
Comment 2: Abstract
The abstract provides an accurate and concise summary of the study, allowing readers to grasp its main objectives and findings. I recommend, however, that the authors modify one of the keywords and include “lipid” as one of them, to better reflect the study’s focus.
Response: The keyword “lipid” was supplemented in the revision.
Comment 3: Introduction
The introduction is well-written and logically organized. Nevertheless, it would benefit from a short background paragraph on marine fungi and their ecological roles in the marine environment, to better contextualize their potential as sources of antifouling compounds.
Response: The distribution and ecological roles of marine fungi, as well as those of their metabolites, were briefly introduced.
Comment 4: Materials and Methods
The methods are clearly described, making it easy to follow the experimental procedures. However, in the GC–MS analysis, the confirmation of some compounds using analytical standards would strengthen the validity of the chemical identifications.
Response: We appreciate your insightful suggestion. Although GC-MS is a well-established method for identifying lipid components, performing validation with standard compounds would significantly enhance the credibility of the chemical identifications. Accordingly, we will incorporate standard compound validation in our future work.
Comment 5: Results
The results are well presented and focused on the most relevant findings.
- InTable 2, the first column is missing a heading; I suggest labeling it as “Fungal lipid fraction” to clarify the distinction among the ten tested fractions.
- In line 177, there seems to be a minor error — the authors refer to four lipid extractions instead of three.
Apart from these minor issues and a few typographical errors, the results section is clearly written and easy to follow.
Response: Thank you for your valuable input. All these errors were revised.
Comment 6: Discussion
The discussion would benefit from a deeper analysis of some key aspects. I recommend expanding this section by incorporating more comparisons with existing literature on antifouling compounds derived from marine microorganisms. This would help strengthen the interpretation of the findings and highlight the novelty of the work.
Response: We thank you for the suggestion. Given that the preceding discussion has elaborated on the advantages of natural antifoulants from marine microorganisms, we have now incorporated a preliminary comparative assessment of the environmental safety profiles between our lipid fractions and SeaNine 211, based on both literature and our experimental data. The results indicate that our lipid components may present a safer profile for environmental organisms.
Comment 7: Conclusions
The conclusions are well supported by the presented results and appropriately summarize the study’s main contributions.
Response: We appreciate your positive feedback.
Additional comments
- Comment 8: Have the authors explored, or do they plan to explore, the molecular mechanisms underlying the antifouling activity of the identified lipids?
Response: We are grateful for your valuable suggestion. Understanding the mechanism of action is critical for the development of natural antifoulants. However, these lipid fractions remain mixtures containing several compounds, which is not conducive to probing their mechanism of action. To address this limitation, we plan to utilize fatty acid standards in our future research to elucidate the underlying mechanisms.
- Comment 9: Given that long-term stability and resistance remain major challenges for antifouling compounds, how do the authors envision addressing the durability of these lipid-based antifouling agents over extended periods (e.g., several years)?
Response: Environmental compatibility and a propensity for rapid biodegradation are recognized advantages of natural antifoulants in ocean. In this study, certain lipid fractions demonstrated excellent anti-fouling efficacy after three months in field experiments. We will conduct prolonged field trials to assess their long-term performance and further evaluate their toxicity on other marine organisms in future work.
Comment 10: In view of the manuscript’s high quality, clarity, and scientific relevance, I recommend it for minor revisions before acceptance.
Response: We would like to express our sincere gratitude for your detailed and constructive review.